# Determinants of stillbirth among women who delivered in hospitals of North Wollo Zone, Northeast Ethiopia: A case-control study

Atnaf Alem Abriham[1], Eyob Shitie[2], Sisay Melese[3], Anteneh Mengist Dessie [4], Asmamaw Demis Bizuneh[5,6]*

1 Gubalafto Health Office, North Wollo Zonal Health Department, Woldia, Amhara Region, Ethiopia, 2 School of Midwifery, College of Health Sciences, Woldia University, Woldia, Ethiopia, 3 CDC Project HIV Case Detection, Linkage, Care and Treatment Coordinator, Woldia, Ethiopia, 4 Department of Public Health, College of Health Sciences, Debre Tabor University, Debre Tabor, Ethiopia, 5 School of Nursing, College of Health Sciences, Woldia University, Woldia, Ethiopia, 6 Monash Center for Health Research and Implementation, Faculty of Medicine, Nursing and Health Sciences, Monash University, Melbourne, Victoria, Australia

* asmamawdemis@gmail.com

## Abstract

### Background

Stillbirth is a silent tragedy that shatters the lives of women, families, and nations. Though affecting over 2 million infants globally in 2019, it remains overlooked, with no specific targets dedicated to its reduction in the sustainable development goals. Insufficient knowledge regarding the primary risk factors contributing to stillbirths hinders efforts to reduce its occurrence. Driven by this urgency, this study focused on identifying the determinants of stillbirth among women giving birth in hospitals across North Wollo Zone, Northeast Ethiopia.

### Methodology

This study employed an institution-based unmatched case-control design, involving a randomly selected sample of 412 women (103 cases and 309 controls) who gave birth in hospitals of North Wollo Zone. Data were collected using a structured data extraction checklist. Data entry was conducted using Epi-data version 3.1, and analysis was performed using SPSS version 25.0. Employing a multivariable logistic regression model, we identified independent predictors of stillbirth. The level of statistical significance was declared at a p-value < 0.05.

### Results

Our analysis revealed several critical factors associated with an increased risk of stillbirth. Women who experienced premature rupture of membranes (AOR = 5.53, 95% CI: 2.33–9.94), induced labor (AOR = 2.24, 95% CI: 1.24–4.07), prolonged labor exceeding 24 hours (AOR = 3.80, 95% CI: 1.94–7.45), absence of partograph monitoring during labor (AOR = 2.45, 95% CI: 1.41–4.26) were all significantly associated with increased risk of stillbirth. Preterm birth (AOR = 3.46, 95% CI: 1.87–6.39), post-term birth (AOR = 3.47, 95% CI: 1.35–

**Funding:** The author(s) received no specific funding for this work.

**Competing interests:** The authors have declared that no competing interests exist.

**Abbreviations:** ANC, Antenatal Care; AOR, Adjusted Odds ratio; APH, Antepartum Hemorrhage; CI, Confidence Interval; COR, Crude Odds Ratio; EDHS, Ethiopian Demographic and Health Survey; PROM, Premature Rapture of Membrane; SDG, Sustainable Development Goals; SVD, Spontaneous Vaginal Delivery; WHO, World Health Organization.

8.91), and carrying a female fetus (AOR = 1.81, 95% CI: 1.02–3.22) were at a higher risk of stillbirth.

## Conclusion

These findings highlight the importance of early intervention and close monitoring for women experiencing premature rupture of membranes, prolonged labor, or induced labor. Additionally, consistent partograph use and enhanced prenatal care for pregnancies at risk of preterm or post-term birth could potentially contribute to reducing stillbirth rates and improving maternal and neonatal outcomes. Further research is needed to investigate the underlying mechanisms behind the observed association between fetal sex and stillbirth risk.

## Background

Fetal death is characterized by the birth of a fetus without any discernible signs of life, such as the absence of breathing, heartbeats, umbilical cord pulsation, or voluntary muscle movement [1]. This event can manifest as either abortions (miscarriages) or stillbirths, contingent on the pregnancy stage and the policy framework of the respective country. The World Health Organization (WHO) has established a globally accepted gestational age for cross-national comparability, defining stillbirth as the delivery of a nonviable fetus, weighing more than 1000 grams, or with a gestation period exceeding 28 full weeks, regardless of whether it occurs before or during labor [2].

Stillbirth remains a heartbreaking reality for numerous families worldwide. In 2019, the global stillbirth rate was 13.9 per 1000 live births, with significant regional variations ranging from 21.7 stillbirths per 1000 total births in Sub-Saharan Africa (SSA) to 2.9 in Western Europe. Overwhelmingly, almost 98% of the stillbirths occurred in low- and middle-income countries (LMICs), with approximately 77% of this burden concentrated in SSA and South East Asian countries [3]. For instance, the 2016 Ethiopian Demographic and Health Survey (EDHS) reported a stillbirth rate of 11.7 per 1000 pregnancies nationally and 23.8 per 1000 in the Amhara region, where this study was conducted [4].

What adds to the tragedy is that a majority of stillbirths result from preventable conditions, including maternal infections (notably syphilis and malaria), non-communicable diseases, and obstetric complications. Although some cases are due to congenital disorders, preventive measures exist for some of these as well [5, 6]. While the cause of stillbirth is not always identifiable (1 in 3 stillbirths cannot be explained), common factors include issues with the placenta and/or umbilical cord, preeclampsia, birth defects, infections, trauma, clotting disorders, and other chronic maternal medical conditions [7].

Stillbirth has significant financial implications for parents and long-term economic repercussions for society [8]. Moreover, stillbirths increase the risk of maternal mortality due to associations with short intervals between pregnancies and inadequate care for women experiencing stillbirth [9, 10]. Despite the severe and enduring effects of stillbirth, stigma, and societal taboos often obscure the hardships faced by families [6].

Regrettably, stillbirth is frequently overlooked and omitted from global data tracking systems, excluding it from critical international initiatives like the Sustainable Development Goals (SDGs) and the Millennium Development Goals (MDGs). This lack of attention poses a

considerable challenge in prioritizing interventions to prevent stillbirths. The civil war in northern Ethiopia has further strained maternal health services, contributing to a higher incidence of stillbirths [11, 12]. In this context, gathering data on the determinants of stillbirths becomes crucial to effectively plan and prioritize maternal and child healthcare services, with a focus on reducing stillbirth rates and improving outcomes. Therefore, this study aimed to identify the determinant factors contributing to stillbirths among women delivering in hospitals within the North Wollo Zone of Northeast Ethiopia.

## Methods

### Study design, area and period

In this unmatched case-control study conducted at a healthcare institution, data were systematically collected from January 1 to January 30, 2023. The study focused on a meticulous examination of medical charts belonging to women who underwent childbirth in hospitals located within the North Wollo Zone, one of the thirteen Zones constituting the Amhara regional state. The administrative hub of this Zone is Woldia, strategically positioned approximately 360 km away from Bahir Dar and 520 km away from Addis Ababa. As per the 2007 census, the population of North Wollo Zone was documented at 1,500,303, with 752,895 men and 747,408 women. Currently, the zone boasts one specialized hospital, one general hospital, four primary hospitals, 69 health centres, and 290 health posts. For this study, data were extracted from the medical records of women who underwent childbirth within the zone.

**Cases.** Women who experienced stillbirth, which is defined as a baby born with no sign of life after 28 weeks gestation or with ≥1000 grams birth weight [13].

**Controls.** Women who gave live births after 28 weeks of gestations [13].

**Partograph utilization.** If all the data on the three components of a partograph (fetal condition, progress of labor, and maternal condition) were completed as per WHO protocol, it was considered that a partograph was utilized [14].

**Cord accidents.** If there was cord prolapse, cord knot, and/or nuchal cord, it was considered that there was a cord accident.

### Study population and eligibility criteria

The study encompassed all women who gave birth to babies in chosen hospitals within the North Wollo Zone from January 1 to December 31, 2022. Nevertheless, charts lacking vital information on key variables like the antenatal period, labor status, and birth status were excluded from the study. Additionally, any charts lost during the data collection process were excluded.

### Sample size and sampling procedure

The sample size for this study was determined using Epi Info version-7 software, employing an unmatched case-control formula. The calculation considered an assumed power of 80%, a significance level of 95%, a case-to-control ratio of 1:3, and the previous study's identification of antenatal care (ANC) follow-up as a significant determinant of stillbirth with a percentage of controls exposed of 74.2% [15]. Consequently, the final sample size was set at 412, comprising 103 cases and 309 controls.

For the selection of cases and controls, a simple random sampling technique was applied. Initially, three hospitals were randomly selected from a total of six hospitals within the North Wollo Zone. Subsequently, a sampling frame was created by compiling the medical record numbers of all births during the study period from the delivery registration book of each

hospital. These births were then categorized as either stillbirths or live births. Finally, the calculated sample size was allocated proportionally to each hospital based on their respective population sizes. From the prepared sampling frame, a simple random sampling technique was used to select the study participants.

## Data collection tool and technique

Data collection for this study utilized a structured checklist that was developed in English. The checklist was formulated following an extensive literature analysis, taking into account insights from prior studies [15], and a review of sample charts to ensure the inclusion of pertinent variables. The checklist encompassed various aspects, including sociodemographic data, obstetric factors, medical history, healthcare-related factors, and birth outcomes.

To aid in collecting data, ten midwives (with bachelor's degrees) were recruited as data collectors, along with three additional midwives (also with bachelor's degrees) who served as supervisors. The data collectors meticulously reviewed the patient's charts, which included information such as medical history, delivery summary, laboratory findings, partograph records, progress notes, and operation notes. Utilizing the checklist, the data collectors extracted the necessary information from these documents and filled out the checklist accordingly.

## Data quality assurance

A comprehensive two-day training was given for data collectors and supervisors on the techniques of data collection and an in-depth explanation of each variable within the data collection charts. The pre-test was conducted on 5% of the sample size to ensure the validity of the tool in Woldia Specialized Hospital. Furthermore, we implemented double data entry by two independent data clerks, followed by a meticulous cross-checking process to verify data consistency. Finally, Cohen's kappa coefficient (k = 0.86) assessed interrater reliability among data collectors, revealing an impressive near-perfect agreement, solidifying the data's validity and trustworthiness.

## Data processing and analysis

The data collected for this study underwent coding, entry, and cleaning processes using Epi-data version 3.1 software. Subsequently, the analysis was conducted using SPSS version 25.0. Categorical variables were summarized through frequency tables and graphs, presenting both numbers and percentages. Conversely, continuous variables were described using measures such as mean/median and standard deviations/interquartile range (IQR), depending on the data distribution.

To evaluate the association between dependent and independent variables, odds ratios with 95% confidence intervals were calculated. Initially, a bi-variable logistic regression model was employed to determine the crude association of the independent variables with the dependent variable. Variables with a p-value less than 0.2 in the bi-variable logistic regression analysis were chosen and included in the multivariable logistic regression model [16]. The multivariable model aimed to control for confounding factors and an adjusted odd ratio (AOR) with 95% CI was estimated to identify the independent predictors of stillbirth. To address multicollinearity, variables exhibiting standard error exceeding two were excluded from the multivariable analysis. Model fitness was evaluated using the Hosmer and Lemeshow test (p = 0.87) suggesting the model was fit. The level of statistical significance was declared at a $p < 0.05$.

## Ethical considerations

Ethical approval was sought from the Institutional Review Board of Woldia University (Protocol No: WDU/IRB001). Support letters were obtained from the North Wollo Zonal Health

Department for each hospital involved. Permission was also obtained from each hospital administrator/manager to access and review the delivery register books and medical cards. To ensure confidentiality, questionnaires were coded instead of using personal identifiers. Data from delivery registry books and charts were securely stored in a locked file cabinet to ensure confidentiality. All procedures were conducted in compliance with the principles outlined in the Declaration of Helsinki.

## Results

### Socio-demographic characteristics

In this study, 412 charts (103 cases and 309 controls) of women delivering in selected North Wollo Zone hospitals were analyzed. The median age of the case women was 32 years (IQR: 34–28), while the control women had a median age of 33 years (IQR: 34–30). Among cases, 11 (10.7%) were below 20 years, compared to 7 (2.3%) in controls. Additionally, 20 (19.4%) of the cases and 60 (19.4%) of the controls were found to be above 35 years.

### Healthcare-related characteristics

The majority, 90.3% of cases and 94.5% of control had at least one ANC visit. However, only 16.5% of cases and 29.1% of controls had four or more ANC visits throughout their pregnancies (Fig 1). Partograph utilization was low among both groups, with only 45.6% of cases and 71.8% of control women having their labor monitored using it.

### Obstetrics and medical-related characteristics

About 87.4% of cases and 93.2% of control women had multiple previous pregnancies. Among them, 93.3% of cases and 76.3% of controls had an interbirth interval of ≤33 months. Concerning the onset of labor, 62.1% of cases and 79.6% of control women experienced

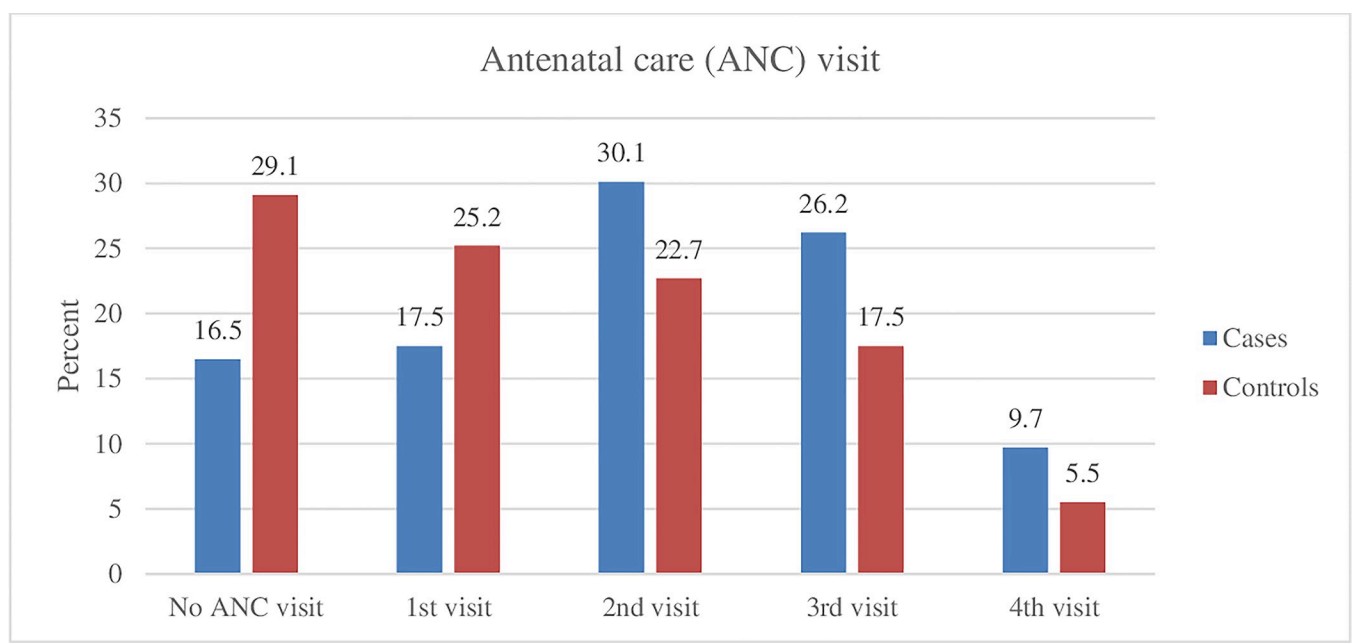

**Fig 1. Antenatal care visit among women who gave birth in North Wollo Zone hospitals, Northeast Ethiopia, 2022.**

**Table 1. Obstetrics and medical-related characteristics of women who gave birth in North Wollo Zone hospitals, Northeast Ethiopia, 2022.**

| Variables | Category | Cases = 103 [n (%)] | Controls = 309 [n (%)] | Total = 412 [n (%)] |
|---|---|---|---|---|
| Gravidity | Primigravida | 13 (12.6) | 21 (6.8) | 34 (8.3) |
| | Multigravida | 79 (76.7) | 284 (91.9) | 363 (88.1) |
| | Grand multigravida | 11 (10.7) | 4 (1.3) | 15 (3.6) |
| History of stillbirth | Yes | 8 (7.8) | 8 (2.6) | 16 (3.9) |
| | No | 95 (92.2) | 301 (97.4) | 396 (96.1) |
| Type of pregnancy | Single | 98 (95.1) | 295 (95.5) | 393 (95.4) |
| | Twin | 5 (4.9) | 14 (4.5) | 19 (4.6) |
| Fetal presentation | Vertex | 90 (87.4) | 278 (90.0) | 368 (89.3) |
| | Not vertex | 13 (11.6) | 31 (10.0) | 44 (10.7) |
| Onset of labor | Induced | 39 (37.9) | 63 (20.4) | 102 (24.8) |
| | Spontaneous | 64 (62.1) | 246 (79.6) | 310 (75.2) |
| Duration of labor | <24 hrs. | 70 (68.0) | 279 (90.3) | 345 (84.6) |
| | ≥24 hrs. | 33 (32.0) | 30 (9.7) | 63 (15.4) |
| PROM | Yes | 22 (21.4) | 12 (3.9) | 34 (8.3) |
| | No | 81 (78.6) | 297 (96.1) | 378 (91.7) |
| Hypertension during pregnancy | Yes | 22 (21.4) | 38 (12.3) | 60 (14.6) |
| | No | 81 (78.6) | 271 (87.7) | 352 (85.4) |
| APH | Yes | 14 (13.6) | 11 (3.6) | 25 (6.1) |
| | No | 89 (86.4) | 298 (96.4) | 387 (93.9) |
| Obstructed labor | Yes | 23 (22.3) | 30 (9.7) | 53 (12.9) |
| | No | 80 (77.7) | 279 (90.3) | 359 (87.1) |
| Cord accident | Yes | 7 (6.8) | 14 (4.5) | 21 (5.1) |
| | No | 96 (93.2) | 295 (95.5) | 391 (94.9) |
| Cord accident type (n = 21) | Cord prolapses | 3 (42.8) | 9 (64.3) | 12 (57.1) |
| | True cord knot | 2 (28.6) | 2 (14.3) | 4 (19.1) |
| | Nuchal cord | 2 (28.6) | 3 (21.4) | 5 (23.8) |
| Mode of delivery | SVD | 64 (62.1) | 194 (62.8) | 258 (62.6) |
| | Instrumental assisted | 25 (24.3) | 88 (28.5) | 113 (27.4) |
| | Cesarean section | 14 (13.6) | 27 (8.7) | 41 (10.0) |
| Chorioamnionitis | Yes | 12 (11.7) | 35 (11.3) | 47 (11.4) |
| | No | 91 (88.3) | 274 (88.7) | 365 (88.6) |
| Malaria | Yes | 3 (2.9) | 5 (1.6) | 8 (1.9) |
| | No | 100 (97.1) | 304 (98.4) | 404 (98.1) |
| Anemia | Yes | 10 (9.7) | 15 (4.9) | 25 (6.1) |
| | No | 93 (90.3) | 294 (95.1) | 387 (93.9) |

APH: antepartum hemorrhage, PROM: premature rupture of membrane, SVD: spontaneous vaginal delivery. Gravidity encompasses three categories: primigravida (one pregnancy history), multigravida (2–4 pregnancy history), and grand multigravida (5 or more pregnancy history).

spontaneous labor. The majority, 91.7% of women did not have premature rupture of membranes (PROM), while only 9% had encountered a cord accident (Table 1).

## Fetal related characteristics

Among the examined fetuses, 38.1% of cases and 5.4% of controls had congenital abnormalities. While only 6% of controls delivered male babies, 20% of cases gave birth to female babies. Both groups primarily delivered within the term window (37–41 weeks), with 49.5% of cases

and 81.6% of controls falling within this timeframe. Additionally, over half (57.3%) of cases weighed between 2500 and 4000 grams compared to a much larger proportion of controls (83.5%) within this range. A total of 13 cases of congenital anomalies were identified in the study among cases (spina bifida = 4, club foot = 3, hydrocephalus = 4, omphalocele = 1, and imperforate anus = 1) and the control group included 14 individuals (spina bifida = 4, club foot = 2, hydrocephalus = 3, omphalocele = 2, imperforate anus = 2, and anencephaly = 1).

## Determinants of stillbirth

Initially, bivariate binary logistic regression analysis was conducted to examine the association between stillbirth and various explanatory variables. Variables significantly associated with a stillbirth at this threshold included gravidity, history of stillbirth, presence of PROM, anaemia, APH, hypertension during pregnancy, sex of the neonate, onset of labor, partograph utilization, gestational age at birth, duration of labor, and congenital malformation. Subsequently, a multivariable logistic regression analysis was performed, including only the significant variables from the bivariate analysis (p-value < 0.2). The significant determinants identified through this analysis were the sex of the fetus, gestational age at birth, presence of PROM, the onset of labor, duration of labor, and partograph utilization.

This study revealed that preterm births had 3.5 times higher odds (AOR = 3.46, 95% CI: 1.87–6.39) and post-term births had 3.5 times higher odds (AOR = 3.47, 95% CI: 1.35–8.91) of being stillbirths compared to term births. Female fetuses were 1.8 times more likely (AOR = 1.81, 95% CI: 1.02–3.22) to result in stillbirths compared to their male counterparts. The odds of experiencing stillbirth were 5.5 times higher (AOR = 5.53, 95% CI: 2.33–9.94) among women with PROM compared to those without PROM. Women who had induced onset of labor and a labor duration of ≥24 hours had 2.2 times higher odds (AOR = 2.24, 95% CI: 1.24–4.07) and 3.8 times higher odds (AOR = 3.80, 95% CI: 1.94–7.45) of having a still-birth, respectively. Furthermore, the odds of experiencing stillbirth were 2.5 times higher (AOR = 2.45, 95% CI: 1.41–4.26) for women whose labor was not followed by a partograph compared to those whose labor was monitored with a partograph (Table 2).

## Discussions

The main aim of this study was to investigate the factors influencing stillbirth among women delivering in North Wollo Zone hospitals in Northeast Ethiopia. The analysis identified several significant determinants of stillbirth, specifically the sex of the fetus, gestational age at birth, presence of premature rupture of membranes (PROM), the onset of labor, duration of labor, and utilization of a partograph.

The findings of this study indicate that both preterm and post-term births were associated with a higher likelihood of stillbirth. These results are consistent with existing studies that have also identified preterm and post-term pregnancies as risk factors for stillbirth [17–19]. The increased risk observed in preterm births may be attributed to respiratory complications caused by the immaturity of the respiratory system [20]. Additionally, premature fetuses may be more vulnerable to ischemia due to incomplete blood-brain barrier development, potentially leading to fetal demise during delivery [21]. Conversely, post-term pregnancies carry a higher risk of stillbirth due to factors such as meconium aspiration syndrome and macrosomia [22]. Post-term pregnancy is also associated with lower umbilical cord pH levels and placental insufficiency, increasing the risk of fetal death before or during delivery [22, 23].

Furthermore, a noteworthy finding of this study was the high prevalence of PROM among women who experienced stillbirth. This result is consistent with previous studies conducted in various settings [24–26], which also demonstrated a significant association between PROM

**Table 2. Determinants of stillbirth among women delivering in North Wollo Zone hospitals, Northeast Ethiopia, 2022.**

| Variables | Category | Stillbirth status | | COR (95%CI) | AOR (95%CI) | p-value |
|-----------|----------|-------------------|---|-------------|-------------|---------|
| | | Cases | Controls | | | |
| Gravidity | | 103 | 309 | 1.32 (1.05–1.65) | 1.09 (0.83–1.42) | 0.567 |
| Sex of newborn | Male | 29 | 148 | 1 | 1 | 0.011 |
| | Female | 74 | 161 | 2.35 (1.45–3.81) | 1.81 (1.02–3.22) | |
| Gestational age at birth | < 37 weeks | 37 | 44 | 4.15 (2.44–7.06) | 3.46 (1.87–6.39) | <0.001 |
| | 37–41 weeks | 51 | 252 | 1 | 1 | 0.011 |
| | ≥ 42 weeks | 15 | 13 | 5.70 (2.56–12.71) | 3.46 (1.87–6.39) | |
| Hypertension during pregnancy | Yes | 22 | 38 | 1.94 (1.08–3.46) | 0.80 (0.37–1.72) | 0.931 |
| | No | 81 | 271 | 1 | 1 | |
| History of stillbirth | Yes | 8 | 8 | 3.17 (1.16–8.67) | 1.57 (0.44–5.67) | 0.537 |
| | No | 95 | 301 | 1 | 1 | |
| Presence of PROM | Yes | 22 | 12 | 6.72 (3.19–14.16) | 5.53 (2.33–9.94) | <0.001 |
| | No | 81 | 297 | 1 | 1 | |
| Presence of APH | Yes | 14 | 11 | 4.26 (1.87–9.72) | 2.48 (0.86–7.12) | 0.066 |
| | No | 89 | 298 | 1 | 1 | |
| Duration of labor | < 24 hrs. | 70 | 279 | 1 | 1 | <0.001 |
| | ≥ 24 hrs. | 33 | 30 | 4.38 (2.51–7.67) | 3.80 (1.94–7.45) | |
| Anemia during pregnancy | Yes | 10 | 15 | 2.12 (0.92–4.85) | 1.40 (0.48–4.08) | 0.525 |
| | No | 93 | 294 | 1 | 1 | |
| Congenital malformation | Yes | 13 | 14 | 3.04 (1.38–6.71) | 1.66 (0.65–4.25) | 0.212 |
| | No | 90 | 295 | 1 | 1 | |
| Onset of labor | Induced | 39 | 63 | 2.38 (1.46–3.86) | 2.24 (1.24–4.07) | 0.015 |
| | Spontaneous | 64 | 246 | 1 | 1 | |
| Partograph use | Yes | 47 | 222 | 1 | 1 | 0.002 |
| | No | 56 | 87 | 3.04 (1.92–4.82) | 2.45 (1.41–4.26) | |

AOR: Adjusted odds ratio, APH: Antepartum hemorrhage, CI: Confidence interval, COR: Crude odds ratio, PROM: Premature rupture of membrane.

and stillbirth. The occurrence of stillbirth in cases of PROM may be attributed to factors such as reduced amniotic fluid, causing oligohydramnios and potential compression of the umbilical cord, leading to fetal hypoxia and fetal demise. Prolonged premature rupture of membranes can also result in ascending uterine infection (chorioamnionitis), contributing to stillbirth [27].

Another significant determinant of stillbirth identified in this study was the sex of the fetus. Surprisingly, the study revealed that the risk of stillbirth was higher in female fetuses compared to male fetuses. This finding contradicts the commonly held assumption that male fetuses are more vulnerable to intrauterine insults and stillbirth [28]. However, this result is supported by a cohort study investigating the association between stillbirth and fetal gender [29]. Although further investigation is warranted, one possible explanation is that carrying a female fetus increases the risk of fetal growth restriction [30].

In addition, this study identified that women with a labor duration of 24 hours or longer were 3.80 times more likely to experience stillbirth. This finding aligns with the results of various studies conducted in Ethiopia [31–33], which consistently reported prolonged labor as a significant risk factor for stillbirth. The prolonged labor may result in fetal distress, leading to intrauterine fetal death. This study also revealed that induced labor onset was also a significant factor associated with an increased risk of stillbirth. This finding is supported by a previous study [31]. The possible explanation for this association is that when labor is induced, the fetus

may be exposed to stressful uterine contractions, which can ultimately lead to fetal demise [34].

Furthermore, the findings of this study demonstrated that women whose labor was not monitored using a partograph had higher odds of experiencing stillbirth compared to those whose labor was monitored with a partograph. This finding is consistent with previous studies [24, 35, 36]. The use of a partograph is crucial as it enables healthcare professionals to identify potential complications during labor and take timely interventions to save the lives of both the mother and the fetus. Without proper monitoring using a partograph, critical factors contributing to stillbirths, such as fetal distress, poor progress in labor, prolonged labor, and obstructed labor, may go unnoticed by healthcare providers [37].

Our study utilized secondary data from hospital medical records, which suffered from data incompleteness, particularly in behavioral factors such as alcohol drinking, cigarette smoking, and nutritional intake during pregnancy. Despite this limitation, our study contributes to the understanding of factors associated with stillbirth in North Wollo Zone hospitals. The findings can be reliably used as a baseline for future prospective studies and may inform potential policy changes.

The implications of this study's findings are significant, highlighting several key factors associated with an increased risk of stillbirth. These identified factors emphasize the importance of targeted interventions and vigilant monitoring during labor, particularly in cases of induced labor and prolonged duration, and close monitoring of pregnancies with the female fetus, PROM, or preterm/post-term deliveries, to reduce the risk of stillbirth. Additionally, the findings may also highlight the need for further research to determine the mechanisms by which these factors increase the risk of stillbirth.

## Conclusion

In general, the findings of this study identified induced labor, prolonged duration of labor, no partograph use, female fetus, premature rupture of membranes (PROM), and preterm and post-term deliveries as key factors significantly raising stillbirth risk. Importantly, most of these factors are modifiable and amenable to interventions. Therefore, the North Wollo Zonal Health Department, hospitals, and other relevant stakeholders must collaborate to embed partograph use in routine care to predict and manage labor progress, which can prevent tragic stillbirths and give every mother and baby the best chance for a healthy birth. Healthcare providers should also provide specialized care to women undergoing induced labor and should document maternal behavioral and lifestyle factors during history-taking and physical examinations. Moreover, future studies should adopt rigorous designs, such as prospective studies, to investigate additional maternal factors encompassing behavioral aspects such as smoking, alcohol consumption, and nutritional considerations. Additionally, investigations into the prospective association of fetal sex with stillbirth are warranted in future studies.

## Supporting information

**S1 Table. Data extraction checklist.**
(PDF)

## Acknowledgments

We express our sincere appreciation to Woldia University for granting us ethical approval to carry out this study. Additionally, we extend our heartfelt gratitude to the dedicated data

collectors and supervisors involved in this research, as their diligent efforts and time commitment were crucial for the successful completion of this study.

## Author Contributions

**Conceptualization:** Atnaf Alem Abriham, Sisay Melese, Asmamaw Demis Bizuneh.

**Data curation:** Atnaf Alem Abriham.

**Formal analysis:** Atnaf Alem Abriham, Eyob Shitie, Sisay Melese, Anteneh Mengist Dessie, Asmamaw Demis Bizuneh.

**Investigation:** Atnaf Alem Abriham.

**Methodology:** Atnaf Alem Abriham, Eyob Shitie, Sisay Melese, Anteneh Mengist Dessie, Asmamaw Demis Bizuneh.

**Software:** Atnaf Alem Abriham, Anteneh Mengist Dessie, Asmamaw Demis Bizuneh.

**Supervision:** Atnaf Alem Abriham, Eyob Shitie, Asmamaw Demis Bizuneh.

**Validation:** Atnaf Alem Abriham, Eyob Shitie, Asmamaw Demis Bizuneh.

**Visualization:** Eyob Shitie, Asmamaw Demis Bizuneh.

**Writing – original draft:** Atnaf Alem Abriham, Eyob Shitie, Sisay Melese, Asmamaw Demis Bizuneh.

**Writing – review & editing:** Atnaf Alem Abriham, Eyob Shitie, Sisay Melese, Anteneh Mengist Dessie, Asmamaw Demis Bizuneh.

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
