## [Decision Letter · Decision Letter 0]

14 Aug 2023

PONE-D-23-22773Determinants of Stillbirth among Women Who Delivered in Hospitals of North Wollo Zone, Northeast Ethiopia: An Unmatched Case-Control StudyPLOS ONE

Dear Dr. Abriham,

Thank you for submitting your manuscript to PLOS ONE. After careful consideration, we feel that it has merit but does not fully meet PLOS ONE’s publication criteria as it currently stands. Therefore, we invite you to submit a revised version of the manuscript that addresses the points raised during the review process.

We look forward to receiving your revised manuscript.

Kind regards,

Mosharop Hossian

Academic Editor

PLOS ONE

Journal Requirements:

https://www.statpearls.com/articlelibrary/viewarticle/23722/?utm_campaign=reviews&utm_content=23722&utm_source=pubmed

https://idswater.com/2020/08/08/what-is-the-chance-of-having-a-stillborn-baby/

https://threadreaderapp.com/hashtag/stillbirth

In your revision ensure you cite all your sources (including your own works), and quote or rephrase any duplicated text outside the methods section. Further consideration is dependent on these concerns being addressed.

6. We are unable to open your Supporting Information file [File Name]. Please kindly revise as necessary and re-upload.

Reviewers' comments:

Reviewer's Responses to Questions

**Comments to the Author**

1. Is the manuscript technically sound, and do the data support the conclusions?

Reviewer #1: Partly

Reviewer #2: Partly

Reviewer #3: Yes

2. Has the statistical analysis been performed appropriately and rigorously? 

Reviewer #1: No

Reviewer #2: Yes

Reviewer #3: Yes

3. Have the authors made all data underlying the findings in their manuscript fully available?

Reviewer #1: Yes

Reviewer #2: Yes

Reviewer #3: Yes

4. Is the manuscript presented in an intelligible fashion and written in standard English?

Reviewer #1: Yes

Reviewer #2: No

Reviewer #3: Yes

5. Review Comments to the Author

Reviewer #1: Thank you for the submission. The article you have presented gives a very good overview of the determinants implicated in stillbirths in the region of North Wollo Zone, Northeast Ethiopia. Overall, the article is publishable with revisions which will contribute to the transparency of the article and clarify some questions about the methodology.

Methods

1. You have mentioned the exclusion of charts with missing/and incomplete information. That is good. But, it is good practice to explicitly state the criteria for which you have excluded specific variables for cases. Perhaps a table or a supplementary file detailing inclusion and exclusion criteria will benefit transparency and reproducibility.

2. Data collection amongst collectors is a point for improvement. How was inter-reliability ensured amongst the data collectors? This could be further elaborated in the methods section

3. Bivariate logistic regression analysis with p < 0.2

o This point is a big point of contention within the statistical community and is not sound practice. The discussion of deciding on factors to be included or excluded based on p value is dubious.

o The point, thereby, is this is not the way to do it – Making multivariate decisions based on univariate analysis.

o I will point you to the following references. Indeed, the authors need to select better methods to produce models.

Huberty, C. J (1989). Problems with stepwise methods: Better alternatives. In B. Thompson (Ed.) Advances in social science methodology (Vol. 1) (pp. 43-70). Greenwich, CT: JAI Press.

Thompson, B. (1995). Stepwise regression and discriminant analysis need not apply here: A guidelines editorial. Educational and Psychological Measurement, 55(4), 525-534.

Discussion

1. There is no discussion of the limitations. What are the limitations of the study? For example, the use of retrospective data, biases in data collection, and study design. Are these not relevant topics to be discussed?

2. There are elements of limited author’s interpretation of the results in the discussion.

a. The authors have done well to convey that the findings are congruent with literature but do not sparsely discuss potential mechanisms for their findings that separate them by area. For instance, “

These results are consistent with existing studies that have also identified preterm and postterm pregnancies as risk factors for stillbirth [21, 31, 37, 38]. The increased risk observed in preterm births may be attributed to respiratory complications caused by the immaturity of the respiratory system [39]. Additionally, premature fetuses may be more vulnerable to ischemia due to incomplete blood-brain barrier development, potentially leading to fetal demise during delivery [40].”

Were the findings greater, less than or have odds ratios compatible with the other articles? Why may that be so? Are the health systems the same? Are the women of the same population. The discussion is very sparse and does not add valuable insight. This is more so a brief explanation of the attributable factors in the results.

3. Future directions.

a. What could the authors do to add on guidance for further studies to enhance/address the limitations within this study so that you offer a more comprehensive understanding to the issue.

4. Could practical implications be discussed within the article? This could greatly improve the discussion given. For example, discussing the results in the scope of literature allows for the credibility of the results. What actions can be taken based on these? Are they also congruent with those in the other studies? Do you suggest anything different based on the given context?

Overall, it is a good start that needs to be further refined before publication. Please do not be discouraged.

Reviewer #2: Background:

"Almost 98% of the stillbirth rate concentrated [Could this be "is/was concentrated"?] among low- and middle-income countries (LMICs).

Even from the LMICs [Could this be "Even among the LMICs"?), about 77% of the burden was in SSA and South East Asian countries [3]"

The 2016 Ethiopian Demographic and Health Survey (EDHS) showed ......, where the study is going to be conducted [4]. [Check the grammar here. Should it be.. "where the study was conducted" instead of going to be conducted?]

However, their grief is often not socially recognized nor fully acknowledged by doctors or society [8-10]. [Should this part refer to healthcare providers in general and not just doctors?]

Stillbirth have financial consequences for parents and long-term economic repercussions for society [11]. [Check grammar]

It will also exacerbate the risk of maternal mortality since stillbirth is associated with short intervals between pregnancies and women still do

not receive appropriate and respectful care when their baby dies during pregnancy or childbirth [12, 13]. [Be consistent in the use of tenses].

Moreover, stillbirths are not specifically targeted in important international initiatives such as the Sustainable Development Goals (SDGs) and the Millennium Development Goals (MDGs). [We are past the MDGs era, so could the grammar be adjusted accordingly? Say something like- stillbirth has neither been specifically targeted in past nor present important int'l initiatives... or something along the lines.

Furthermore, the civil war in northern Ethiopia has had a detrimental impact on maternal health

services, leading to a high incidence of stillbirths. Given these circumstances, it becomes crucial

to gather data on the determinants of stillbirths in order to plan and prioritize maternal and child

healthcare services effectively, aiming to reduce the frequency and consequences of stillbirths.

Unfortunately, there is limited information available on the determinants of stillbirths specific to

the study area. [There has been no prior reference to Ethiopian context in the manuscript before jumping on to this specific sentence about the country. Add some contextual information about the country first].

Therefore, this study aimed to identify the determinant factors that contribute to stillbirths among women who delivered in hospitals within the North Wollo Zone of Northeast Ethiopia. [There is a need for more contextual information on why this area of the country was chosen for this study- is this region more disadvantaged than the rest of the country? Or something else...?]

Methods:

In this unmatched case-control study conducted at a healthcare institution, data were collected over a period of one month, specifically from January 1 to January 30, 2023. [Some tightening in the writing needed here.. Could say something like - "In this...institution, data collection was carried out over a period of one month from January 1-30, 2023...." or something along the lines].

However, charts with missing or incomplete information regarding key variables.. [information 'on/about' key variables?]

Controls: Women who gave live births after 28th weeks of gestations: [28 weeks of gestation?]

Cord accidents: If there was cord prolapse, cord knot, and/or nuchal cord, it was considered that there was a cord accident in this study. [Can "in this study" be removed?]

Data collection for this study was conducted using a structured checklist that was developed in the English language... [in English.]

with three degree holder midwives.. Not clear what this means. Master's degree?

Results:

Regarding the residence, 252 (61.2%) of the study subjects were rural dwellers, with 63 (25.0%) being cases and 189 (75.0%) being controls. [The first 3 words can be removed- the writing in this manuscript needs tightening].

In terms of partograph use, partograph was used in 47 (45.6%) of the case women' labor and 222 (71.8%) of the control women' labor. [Do the writers mean 'women's..."?]

Figure 1: The spelling of "controls" in the index needs to be corrected to "controls"

About 90 (87.4%) of the cases and 288 (93.2%) of the control women had previously been pregnant multiple times. [If saying 90 women specifically, best to avoid the use of "about"]

Table 1- type of pregnancy- spelling of twins needs to be corrected.

Be consistent on how you write p-value and stick to one format.

Table 2: P in Pregnancy does not need capitalisation.

Discussion:

Although further investigation is warranted, one possible explanation is that carrying a female fetus increases the risk of fetal growth restriction... Could this be elaborated a little as this is an unusual finding?

Conclusion:

Importantly, many of these factors are modifiable and amenable to interventions. " Which one of these are modifiable and amenable to interventions and what could be some examples of influencing/modifying these factors? Some contextual elaboration would be helpful"

Therefore, it is crucial for the Ethiopian Ministry of Health and other relevant stakeholders to prioritize strengthening the utilization of partographs and establishing a robust referral system. "Which of the issues identified by the study will be addressed by robust referrals?"

Overall comments:

Background: This section needs more contextual information on Ethiopia and the region where the study has been conducted. Fix grammatical errors.

Methods: Information on whether multicollinearity between variables was assessed has not been provided. Was multicollinearity assessed? Would also be helpful to see in more detail how the variables were chosen for this study.

Reviewer #3: Determinants of Stillbirth among Women Who Delivered in Hospitals of North Wollo Zone, Northeast Ethiopia: An Unmatched Case-Control Study

Title of study:

The title of study seems very big.

Abstract:

Remove this line: “Factors such as premature rupture of membranes, induced onset of labor, labor duration of 24 hours or more, absence of partograph use during labor, preterm birth, postterm birth, and carrying a female fetus were found to be significantly associated with a higher risk of stillbirth.” Because it is already presented in results of abstract section.

Introduction:

You can write no before heartbeat, no pulsation, and no definite movements of voluntary muscles in the following sentence absent breathing, heartbeats, pulsation of the umbilical cord, or definite movements of voluntary muscles.

The background information regarding the still birth is very comprehensive and presented in very clear language. I feel that your background lacks citations, especially the last four paragraphs of your introduction section. Additionally, I would suggest you remove the content related to the psychological well being and financial spending, because I did not find anything in other sections. Moreover, you need to summarize your introduction to maximum 600-700 words.

Methods:

Keep a separate heading for study design. In the first line of your study design heading, you need to show you have retrospectively analysed one year data, of which you classified X as control and Y as cases. Because this is the unmatched case-control study, so you need to define your cases and control.

Write full forms of ANC.

How you selected 3 hospital randomly? I am interested to know further.

How birth were classified as still birth and live birth? You need to present it with proper citations.

Remove operation definition and terms and all the contents. Rather present all the terminologies and definition within the text, where/if needed. For example: You need to present definition of cases and control of your study in the study design section.

Can you please put your checklist in the supplementary file.

In the main analysis of your method section, VIF and multicollinearity is not discussed. However, in the abstract methodology you presented.

Results:

I read your complete results and I found that everywhere, you are writing the percentage of cases and control. For descriptive statistics, you need to present sample characteristics rather than of each subgroup. However, for presenting the difference between sample characteristics you need to show statistically using some inferential statistics. I am requesting you delete all the descriptive, where you are comparing X% control and X% cases. Rather present whole sample characteristics and write specifically for control and cases if statistical difference between two groups exist (highly recommended changes).

Table 1: Write full form of APH, PROM and SVD in the footnote of table 1.

What are the congenital anormalities observed. You need to write % of all congenital anormalities observed.

Discussion:

Very clear and concise. Seek help from the author for writing introduction.

Also write strengths limitation of your study before conclusion.

I am also interested to know about the implications of this study.

6. PLOS authors have the option to publish the peer review history of their article (what does this mean?). If published, this will include your full peer review and any attached files.

Reviewer #1: No

Reviewer #2: **Yes: **Rupesh Gautam

Reviewer #3: **Yes: **Asif Khaliq

---

## [Author Response · Author response to Decision Letter 0]

17 Jan 2024

Manuscript ID number: PONE-D-23-22773

Title of paper: Determinants of Stillbirth among Women Who Delivered in Hospitals of North Wollo Zone, Northeast Ethiopia: A Case-Control Study.

Point-by-point Response for the Editor and reviewers

First, we would like to thank you for allowing us to submit a revised draft of the manuscript “Determinants of Stillbirth among Women Who Delivered in Hospitals of North Wollo Zone, Northeast Ethiopia: An Unmatched Case-Control Study” for publication in PLOS ONE journal. You [Mosharop Hossian] have requested us to provide a response letter addressing the comment forwarded by the reviewers when submitting the revised manuscript. We appreciate the time and effort that you and the reviewers dedicated to providing feedback on our manuscript and are grateful for the insightful comments on and valuable improvements to our paper. We have incorporated all the suggestions made by the reviewers. Those changes are highlighted within the manuscript. Please see below, for a point-by-point response to the reviewers’ comments and concerns. All page numbers refer to the revised manuscript file with tracked changes.

Academic Editor:

Author response: Thank you for your constructive suggestions and comments; we revised the manuscript as per the PLOS ONE publication style and requirements.

2) We noticed you have some minor occurrence of overlapping text with the following previous publication(s), which needs to be addressed? 

Author response: Thank you for your constructive suggestions; we revised all sections of the manuscript and the current version is free of any overlapping from any previously published articles.

3) Please provide additional details regarding participant consent. In the ethics statement in the Methods and online submission information, please ensure that you have specified (1) whether consent was informed and (2) what type you obtained (for instance, written or verbal, and if verbal, how it was documented and witnessed). If your study included minors, state whether you obtained consent from parents or guardians. If the need for consent was waived by the ethics committee, please include this information. 

Author response: Thank you for your constructive suggestions; we used secondary data analysis and an ethical approval was sought form Institutional Review Board of Woldia University (Protocol No: WDU/IRB001). Support letters were obtained from the North Wollo Zonal Health Department for each hospital involved. Permission was also obtained from each hospital to access and review the delivery register books and medical cards [Refer to Ethical consideration section Page 7 line 164-172].

4) PLOS requires an ORCID iD for the corresponding author in Editorial Manager on papers submitted after December 6th, 2016. Please ensure that you have an ORCID iD and that it is validated in Editorial Manager.

Author response: Thank you for your constructive suggestions; In response, the last author (Asmamaw Demis Bizuneh) has thoroughly reviewed and revised the relevant section of the manuscript. Consequently, he will be included as a corresponding author, sharing equal status with the primary author, and ORCID-ID has been provided.

5) Your ethics statement should only appear in the Methods section of your manuscript. If your ethics statement is written in any section besides the Methods, please move it to the Methods section and delete it from any other section. Please ensure that your ethics statement is included in your manuscript, as the ethics statement entered into the online submission form will not be published alongside your manuscript.

Author response: Thank you for your constructive suggestions; we added the ethics statement under the method section [Refer to Ethical consideration section Page 7 line 164-172].

6) We are unable to open your Supporting Information file [File Name]. Please kindly revise as necessary and re-upload.

Author response: Thank you for your constructive suggestions; the primary author (Atnaf Alem) mistakenly uploaded the Stata file in the first submission and in the revised section we uploaded supplementary tables 1 and 2.

Point-by-point Response for Reviewer-1 Evaluation

First of all, we would like to express our deepest gratefulness and thanks for providing us with such constructive comments and suggestions. We have accepted your comments and acknowledged your devoted efforts to modify the manuscript. Conclusively, you have raised some issues and we have tried to reply to your concerns below.

Methods

1) You have mentioned the exclusion of charts with missing/and incomplete information. That is good. But, it is good practice to explicitly state the criteria for which you have excluded specific variables for cases. Perhaps a table or a supplementary file detailing inclusion and exclusion criteria will benefit transparency and reproducibility. 

Author response: Thank you for your constructive suggestions and comments; included your comments and a supplementary file containing inclusion and exclusion criteria has been provided section [Refer to method section page 5 lines 107-111].

2) Data collection amongst collectors is a point for improvement. How was inter-reliability ensured amongst the data collectors? This could be further elaborated in the methods section. 

Author response: Thank you for your constructive suggestions and comments; we assessed the inter-reliability of data collectors using Cohen’s kappa coefficient (k=0.86) showing almost a perfect agreement [Refer method section page 6 lines 139-146]. 

3) Bivariate logistic regression analysis with p < 0.2. This point is a big point of contention within the statistical community and is not sound practice. The discussion of deciding on factors to be included or excluded based on p value is dubious. The point, thereby, is this is not the way to do it – Making multivariate decisions based on univariate analysis. I will point you to the following references. Indeed, the authors need to select better methods to produce models.

Author response: Thank you for your constructive suggestions and comments; we find the attached references very useful. However, since we have too many covariates in binary logistic regression model, we decided to include those variables with p-value <0.2 for multiple logistic regression analysis. In this regard, different analysts use different techniques, but in most published articles variables having p <0.2 have been used as a cut-off value for multiple logistic regression and we also decided on this regard.

Discussion

1) There is no discussion of the limitations. What are the limitations of the study? For example, the use of retrospective data, biases in data collection, and study design. Are these not relevant topics to be discussed?

Author response: Thank you for your constructive suggestions and comments; We added the limitations of the study in the discussion section as per your suggestions [Refer to discussion section page 12 lines 278-283].

2) There are elements of limited author’s interpretation of the results in the discussion. The authors have done well to convey that the findings are congruent with literature but do not sparsely discuss potential mechanisms for their findings that separate them by area. For instance. Were the findings greater, less than or have odds ratios compatible with the other articles? Why may that be so? Are the health systems the same? Are the women of the same population? The discussion is very sparse and does not add valuable insight. This is more so a brief explanation of the attributable factors in the results.

Author response: Thank you for your constructive suggestions and comments; we tried to discuss how those identifiable risk factors were scientifically plausible and our discussion points have been put in the discussion [Refer to the discussion section page 11 lines 238-3].

3) Future directions. a. What could the authors do to add guidance for further studies to enhance/address the limitations within this study so that you offer a more comprehensive understanding to the issue?

Author response: Thank you for your constructive suggestions and comments; we added the future direction section by highlighting future studies should address the limitation that has occurred in our study [Refer to discussion and conclusion section page 13 lines 284-304].

4) Could practical implications be discussed within the article? This could greatly improve the discussion given. For example, discussing the results in the scope of literature allows for the credibility of the results. What actions can be taken based on these? Are they also congruent with those in the other studies? Do you suggest anything different based on the given context?

Author response: Thank you for your constructive suggestions and comments; we addressed all the given suggestions and we suggest the responsible bodies to minimize the magnitude of stillbirth based on the results of the study [Refer to discussion section page 13 lines 284-290].

Point-by-point Response for Reviewer-2 Evaluation

First of all, we would like to express our deepest gratefulness and thanks for providing us with such constructive comments and suggestions. We have tried to reply to your concerns below.

1) Background:

"Almost 98% of the stillbirth rate concentrated [Could this be "is/was concentrated"?] among low- and middle-income countries (LMICs).

Even from the LMICs [Could this be "Even among the LMICs"?), about 77% of the burden was in SSA and South East Asian countries [3]"

The 2016 Ethiopian Demographic and Health Survey (EDHS) showed ......, where the study is going to be conducted [4]. [Check the grammar here. Should it be.. "where the study was conducted" instead of going to be conducted?]

However, their grief is often not socially recognized nor fully acknowledged by doctors or society [8-10]. [Should this part refer to healthcare providers in general and not just doctors?]

Stillbirth have financial consequences for parents and long-term economic repercussions for society [11]. [Check grammar]

It will also exacerbate the risk of maternal mortality since stillbirth is associated with short intervals between pregnancies and women still do

not receive appropriate and respectful care when their baby dies during pregnancy or childbirth [12, 13]. [Be consistent in the use of tenses].

Moreover, stillbirths are not specifically targeted in important international initiatives such as the Sustainable Development Goals (SDGs) and the Millennium Development Goals (MDGs). [We are past the MDGs era, so could the grammar be adjusted accordingly? Say something like- stillbirth has neither been specifically targeted in past nor present important int'l initiatives... or something along the lines.

Furthermore, the civil war in northern Ethiopia has had a detrimental impact on maternal health

services, leading to a high incidence of stillbirths. Given these circumstances, it becomes crucial

to gather data on the determinants of stillbirths in order to plan and prioritize maternal and child

healthcare services effectively, aiming to reduce the frequency and consequences of stillbirths.

Unfortunately, there is limited information available on the determinants of stillbirths specific to the study area. [There has been no prior reference to Ethiopian context in the manuscript before jumping on to this specific sentence about the country. Add some contextual information about the country first].

Therefore, this study aimed to identify the determinant factors that contribute to stillbirths among women who delivered in hospitals within the North Wollo Zone of Northeast Ethiopia. [There is a need for more contextual information on why this area of the country was chosen for this study- is this region more disadvantaged than the rest of the country? Or something else...?] 

Author response: Thank you very much for your constructive suggestions and comments; we accepted your comments and incorporated them under the background section of the manuscript. We addressed all the grammar-related issues under the whole section of the manuscript [Refer to background section page 3 lines 50-86]. 

2) Methods:

In this unmatched case-control study conducted at a healthcare institution, data were collected over a period of one month, specifically from January 1 to January 30, 2023. [Some tightening in the writing needed here.. Could say something like - "In this...institution, data collection was carried out over a period of one month from January 1-30, 2023...." or something along the lines]. However, charts with missing or incomplete information regarding key variables.. [information 'on/about' key variables?]

Controls: Women who gave live births after 28th weeks of gestations: [28 weeks of gestation?]

Cord accidents: If there was cord prolapse, cord knot, and/or nuchal cord, it was considered that there was a cord accident in this study. [Can "in this study" be removed?]

Data collection for this study was conducted using a structured checklist that was developed in the English language... [in English.] with three degree holder midwives.. Not clear what this means. Master's degree?

Author response: Thank you very much for your constructive suggestions and comments; we accepted your comments and incorporated them under the method section of the manuscript. [Refer to method section page 4 lines 89-172]. 

3) Results: Regarding the residence, 252 (61.2%) of the study subjects were rural dwellers, with 63 (25.0%) being cases and 189 (75.0%) being controls. [The first 3 words can be removed- the writing in this manuscript needs tightening].

In terms of partograph use, partograph was used in 47 (45.6%) of the case women' labor and 222 (71.8%) of the control women' labor. [Do the writers mean 'women's..."?] Figure 1: The spelling of "controls" in the index needs to be corrected to "controls". About 90 (87.4%) of the cases and 288 (93.2%) of the control women had previously been pregnant multiple times. [If saying 90 women specifically, best to avoid the use of "about"]

Table 1- type of pregnancy- spelling of twins needs to be corrected.

Be consistent on how you write p-value and stick to one format.

Table 2: P in Pregnancy does not need capitalisation.

Author response: Thank you very much for your constructive suggestions and comments; we accepted your comments and incorporated them under the result section of the manuscript. [Refer to Result section page 7- 11 lines 174-230]. 

4) Discussion:Although further investigation is warranted, one possible explanation is that carrying a female fetus increases the risk of fetal growth restriction... Could this be elaborated a little as this is an unusual finding?

Author response: Thank you very much for your constructive suggestions and comments; we accepted your comments and incorporated them under the discussion section of the manuscript. [Refer to discussion section page 12 lines 255-261]. 

5) Conclusion: Importantly, many of these factors are modifiable and amenable to interventions. " Which one of these are modifiable and amenable to interventions and what could be some examples of influencing/modifying these factors? Some contextual elaboration would be helpful". Therefore, it is crucial for the Ethiopian Ministry of Health and other relevant stakeholders to prioritize strengthening the utilization of partographs and establishing a robust referral system. "Which of the issues identified by the study will be addressed by robust referrals?"

Author response: Thank you very much for your constructive suggestions and comments; we accepted your comments and incorporated them under the conclusion section of the manuscript. [Refer to conclusion section page 13 lines 291-304]. 

Overall comments:

6) Background: This section needs more contextual information on Ethiopia and the region where the study has been conducted. Fix grammatical errors.

Author response: Thank you very much for your constructive suggestions and comments; we accepted your comments and incorporated them under the background section of the manuscript. We addressed all the grammar-related issues under the whole section of the manuscript [Refer to background section page 3 lines 50-86]. 

7) Methods: Information on whether multicollinearity between variables was assessed has not been provided. Was multicollinearity assessed? Would also be helpful to see in more detail how the variables were chosen for this study.

Author response: Thank you very much for your constructive suggestions and comments; we accepted your comments and incorporated them under the method section of the manuscript. [Refer to method section page 4 lines 89-172]. 

Point-by-point Response for Reviewer-3 Evaluation

First of all, we would like to express our deepest gratefulness and thanks for providing us with such constructive comments and suggestions. We have tried to reply to your concerns below.

1) Abstract: Remove this line: “Factors such as premature rupture of membranes, induced onset of labor, labor duration of 24 hours or more, absence of partograph use during labor, preterm birth, postterm birth, and carrying a female fetus were found to be significantly associated with a higher risk of stillbirth.” Because it is already presented in results of abstract section.

Author response: Thank you very much for your constructive suggestions and comments; we accepted your comments and incorporated them under the abstract section of the manuscript. [Refer to abstract section page 2 lines 23-47]. 

2) Introduction: You can write no before heartbeat, no pulsation, and no definite movements of voluntary muscles in the following sentence absent breathing, heartbeats, pulsation of the umbilical cord, or definite movements of voluntary muscles.

Author response: Thank you very much for your constructive suggestions and comments; we accepted your comments and incorporated them under the background section of the manuscript [Refer background section page 3 lines 50-86]. 

3) The background information regarding the still birth is very comprehensive and presented in very clear language. I feel that your background lacks citations, especially the last four paragraphs of your introduction section. Additionally, I would suggest you remove the content related to the psychological well being and financial spending, because I did not find anything in other sections. Moreover, you need to summarize your introduction to maximum 600-700 words.

Author response: Thank you very much for your constructive suggestions and comments; we accepted your comments and incorporated them under the background section of the manuscript [Refer background section page 3 lines 50-86]. 

4) Methods: Keep a separate heading for study design. In the first line of your study design heading, you need to show you have retrospectively analysed one year data, of which you classified X as control and Y as cases. Because this is the unmatched case-control study, so you need to define your cases and control.

Author response: Thank you very much for your constructive suggestions and comments; we found your comments very useful and we tried to follow the PLOS ONE submission guidelines [Refer to method section page 4 lines 88-126]. 

5) Write full forms of ANC.

Author response: Thank you very much for your constructive suggestions and comments; we accepted your comments and incorporated it. 

6) How you selected 3 hospital randomly? I am interested to know further.

Author response: Thank you very much for your constructive suggestions and comments; out of the six hospitals in the North Wollo zone, we selected three hospitals randomly and then the data were collected from the three hospitals [Refer to method section page 5 lines 105-117]. 

7) How birth were classified as still birth and live birth? You need to present it with proper citations.

Author response: Thank you very much for your constructive suggestions and comments; we accepted your comments and suggestions [Refer to background section page 3]. 

8) Remove operation definition and terms and all the contents. Rather present all the terminologies and definition within the text, where/if needed. For example: You need to present definition of cases and control of your study in the study design section.

Author response: Thank you very much for your constructive suggestions and comments; we accepted your comments and amended them as per your comments [Refer to method section page 4 lines 99-104]. 

9) Can you please put your checklist in the supplementary file.

Author response: Thank you very much for your constructive suggestions and comments; we have incorporated the checklist in the supplementary file [Refer to Supplementary file]. 

10) In the main analysis of your method section, VIF and multicollinearity is not discussed. However, in the abstract methodology you presented.

Author response: Thank you very much for your constructive suggestions and comments; we incorporated them in the method section of the main body of the manuscript [Refer to the method section on page 6 lines 148-163]. 

11) Results:I read your complete results and I found that everywhere, you are writing the percentage of cases and control. For descriptive statistics, you need to present sample characteristics rather than of each subgroup. However, for presenting the difference between sample characteristics you need to show statistically using some inferential statistics. I am requesting you delete all the descriptive, where you are comparing X% control and X% cases. Rather present whole sample characteristics and write specifically for control and cases if statistical difference between two groups exist (highly recommended changes).

Author response: Thank you very much for your constructive suggestions and comments; we accepted your comments and incorporated them under the result section of the manuscript [Refer to result section]. 

12) Table 1: Write full form of APH, PROM and SVD in the footnote of table 1.

Author response: Thank you very much for your constructive suggestions and comments; we accepted your comments and incorporated them under the footnote of table 1 [Refer Table 1]. 

13) What are the congenital anormalities observed? You need to write % of all congenital anormalities observed.

Author response: Thank you very much for your constructive suggestions and comments; we included results describing the types of congenital anomalies in the result section [Refer to result section page 9 lines 196-205]. 

14) Discussion: Very clear and concise. Seek help from the author for writing introduction.

Author response: Thank you very much. 

15) Also write strengths limitation of your study before conclusion.

Author response: Thank you very much for your constructive suggestions and comments; we included them in the discussion of the main manuscript [Refer to the discussion section page 12 lines 278-283]. 

16) I am also interested to know about the implications of this study.

Author response: Thank you very much for your constructive suggestions and comments; we described clearly the implications of the study [Refer to Discussion section page 5 line1-6]. 

We thank you!

---

## [Decision Letter · Decision Letter 1]

29 Jan 2024

PONE-D-23-22773R1Determinants of Stillbirth among Women Who Delivered in Hospitals of North Wollo Zone, Northeast Ethiopia: A Case-Control Study.PLOS ONE

Dear Dr. Bizuneh,

Thank you for submitting your manuscript to PLOS ONE. After careful consideration, we feel that it has merit but does not fully meet PLOS ONE’s publication criteria as it currently stands. Therefore, we invite you to submit a revised version of the manuscript that addresses the points raised during the review process.

We look forward to receiving your revised manuscript.

Kind regards,

Abera Mersha, MSc.

Academic Editor

PLOS ONE

Journal Requirements:

Reviewers' comments:

Reviewer's Responses to Questions

**Comments to the Author**

1. If the authors have adequately addressed your comments raised in a previous round of review and you feel that this manuscript is now acceptable for publication, you may indicate that here to bypass the “Comments to the Author” section, enter your conflict of interest statement in the “Confidential to Editor” section, and submit your "Accept" recommendation.

Reviewer #1: All comments have been addressed

Reviewer #2: All comments have been addressed

2. Is the manuscript technically sound, and do the data support the conclusions?

Reviewer #1: Yes

Reviewer #2: Yes

3. Has the statistical analysis been performed appropriately and rigorously? 

Reviewer #1: Yes

Reviewer #2: Yes

4. Have the authors made all data underlying the findings in their manuscript fully available?

Reviewer #1: Yes

Reviewer #2: Yes

5. Is the manuscript presented in an intelligible fashion and written in standard English?

Reviewer #1: Yes

Reviewer #2: No

6. Review Comments to the Author

Reviewer #1: No further comments. The work is rigorous enough for publication.

Reviewer #2: Thank you for the opportunity to have a look at the manuscript once again. The authors have made most changes pointed out in the first round.

The findings also provide strength to the argument that if the identified (and amendable) interventions are implemented by the local health authorities, it might help bring down the total number of preventable stillbirths. So, the study does have merits in the local context (as well as comparable socioeconomic and geographic contexts outside Ethiopia) from an implementation point of view.

Minor corrections (I do not need to review these changes again):

Fix typos in line numbers 40 (Prolonged should be prolonged), 193, 240, 280.

Line 104: 28 weeks and not 28th weeks, to be consistent with the definition of cases.

Line 111: Gave birth to babies, instead of delivered babies?

Add a note defining gravidity categories under table 1.

Line 200 reads awkward (male babies vs girls- use consistent terminology).

7. PLOS authors have the option to publish the peer review history of their article (what does this mean?). If published, this will include your full peer review and any attached files.

Reviewer #1: No

Reviewer #2: No

---

## [Author Response · Author response to Decision Letter 1]

13 Mar 2024

Manuscript ID number: PONE-D-23-22773R1

Title of paper: Determinants of Stillbirth among Women Who Delivered in Hospitals of North Wollo Zone, Northeast Ethiopia: A Case-Control Study.

Point-by-point Response for the Editor and reviewers

First, we would like to thank you for allowing us to submit the second revised draft of the manuscript “Determinants of Stillbirth among Women Who Delivered in Hospitals of North Wollo Zone, Northeast Ethiopia: An Unmatched Case-Control Study” for publication in PLOS ONE journal. You [Abera Mersha] have requested us to provide a response letter addressing the comment forwarded by the reviewers when submitting the revised manuscript. We appreciate the time and effort that you and the reviewers dedicated to providing feedback on our manuscript and are grateful for the insightful comments on and valuable improvements to our paper. We have incorporated all the suggestions made by the reviewers. Those changes are highlighted within the manuscript. Please see below, for a point-by-point response to the reviewers’ comments and concerns.

Academic Editor:

a. A rebuttal letter that responds to each point raised by the academic editor and reviewer(s). 

b. A marked-up copy of your manuscript that highlights changes made to the original version. 

c. An unmarked version of your revised paper without tracked changes.

Author response: Thank you. We included all the requested documents while submitting the revised manuscript. 

Point-by-point Response for Reviewer-2 Evaluation

First of all, we would like to express our deepest gratefulness and thanks for providing us with such constructive comments and suggestions. We have accepted your comments and acknowledged your devoted efforts to modify the manuscript.

1) The findings also provide strength to the argument that if the identified (and amendable) interventions are implemented by the local health authorities, it might help bring down the total number of preventable stillbirths. So, the study does have merits in the local context (as well as comparable socioeconomic and geographic contexts outside Ethiopia) from an implementation point of view.

Author response: Thank you for your suggestions; it gives good insight for local authorities to implement in their strategic plan to minimize preventable still birth.

2) Minor corrections (I do not need to review these changes again):

a. Fix typos in line numbers 40 (Prolonged should be prolonged), 193, 240, 280.

b. Line 104: 28 weeks and not 28th weeks, to be consistent with the definition of cases.

c. Line 111: Gave birth to babies, instead of delivered babies?

d. Add a note defining gravidity categories under table 1.

e. Line 200 reads awkward (male babies vs girls- use consistent terminology).

Author response: Thank you for your constructive suggestions and comments; we addressed all typo corrections in the whole section of the manuscript accordingly. 

We thank you!

---

## [Decision Letter · Decision Letter 2]

20 Mar 2024

Determinants of Stillbirth among Women Who Delivered in Hospitals of North Wollo Zone, Northeast Ethiopia: A Case-Control Study.

PONE-D-23-22773R2

Dear Dr. Bizuneh,

We’re pleased to inform you that your manuscript has been judged scientifically suitable for publication and will be formally accepted for publication once it meets all outstanding technical requirements.

An invoice for payment will follow shortly after the formal acceptance. To ensure an efficient process, please log into Editorial Manager at Editorial Manager® , click the 'Update My Information' link at the top of the page, and double check that your user information is up-to-date. If you have any billing related questions, please contact our Author Billing department directly at authorbilling@plos.org.

Kind regards,

Abera Mersha, MSc.

Academic Editor

PLOS ONE

Additional Editor Comments (optional):

Reviewers' comments:

Reviewer's Responses to Questions

**Comments to the Author**

1. If the authors have adequately addressed your comments raised in a previous round of review and you feel that this manuscript is now acceptable for publication, you may indicate that here to bypass the “Comments to the Author” section, enter your conflict of interest statement in the “Confidential to Editor” section, and submit your "Accept" recommendation.

Reviewer #2: All comments have been addressed

2. Is the manuscript technically sound, and do the data support the conclusions?

Reviewer #2: Yes

3. Has the statistical analysis been performed appropriately and rigorously? 

Reviewer #2: I Don't Know

4. Have the authors made all data underlying the findings in their manuscript fully available?

Reviewer #2: Yes

5. Is the manuscript presented in an intelligible fashion and written in standard English?

Reviewer #2: Yes

6. Review Comments to the Author

Reviewer #2: Thank you for making further changes in the manuscript in line with the reviewers' comments. We hope the findings of this paper will help the local health authorities in their planning.

7. PLOS authors have the option to publish the peer review history of their article (what does this mean?). If published, this will include your full peer review and any attached files.

Reviewer #2: No

---

## [Editor Report · Acceptance letter]

1 Apr 2024

PONE-D-23-22773R2 

PLOS ONE

Dear Dr. Bizuneh, 

I'm pleased to inform you that your manuscript has been deemed suitable for publication in PLOS ONE. Congratulations! Your manuscript is now being handed over to our production team.

Kind regards, 

on behalf of

Mr. Abera Mersha 

Academic Editor

PLOS ONE